# Hypoalbuminemia Reflects Nutritional Risk, Body Composition and Systemic Inflammation and Is Independently Associated with Survival in Patients with Colorectal Cancer

**DOI:** 10.3390/cancers12071986

**Published:** 2020-07-21

**Authors:** Arwa S. Almasaudi, Ross D. Dolan, Christine A. Edwards, Donald C. McMillan

**Affiliations:** 1Department of Clinical Nutrition, Faculty of Applied Medical Science, King Abdulaziz University, Jeddah 21589, Saudi Arabia; 2Human Nutrition, School of Medicine, Dentistry and Nursing, College of Medical, Veterinary and Life of Sciences, University of Glasgow, Glasgow Royal Infirmary, Glasgow G31 2ER, UK; Christine.edwards@glasgow.ac.uk; 3Academic Unit of Surgery, College of Medical, Veterinary and Life of Sciences-University of Glasgow, Royal Infirmary, Glasgow G31 2ER, UK; Ross.Dolan@glasgow.ac.uk (R.D.D.); Donald.McMillan@glasgow.ac.uk (D.C.M.)

**Keywords:** colorectal cancer, albumin, body composition, nutrition status, systemic inflammatory response and colorectal cancer outcomes

## Abstract

It has long been recognized that albumin has prognostic value in patients with cancer. However, although the Global Leadership Initiative on Malnutrition GLIM criteria (based on five diagnostic criteria, three phenotypic criteria and two etiologic criteria) recognize inflammation as an important etiologic factor in malnutrition, there are limited data regarding the association between albumin, nutritional risk, body composition and systemic inflammation, and whether albumin is associated with mortality independent of these parameters. The aim of this study was to examine the relationship between albumin, nutritional risk, body composition, systemic inflammation, and outcomes in patients with colorectal cancer (CRC). A retrospective cohort study (*n* = 795) was carried out in which patients were divided into normal and hypoalbuminaemic groups (albumin  < 35 g/L) in the presence and absence of a systemic inflammatory response C-reactive protein (CRP > 10 and <10 mg/L, respectively). Post-operative complications, severity of complications and mortality were considered as outcome measures. Categorical variables were analyzed using Chi-square test χ^2^ or linear-by-linear association. Survival data were analyzed using univariate and multivariate Cox regression. In the presence of a systemic inflammatory response, hypoalbuminemia was directly associated with Malnutrition Universal Screening Tool MUST (*p* < 0.001) and inversely associated with Body Mass Index BMI (*p* < 0.001), subcutaneous adiposity (*p* < 0.01), visceral obesity (*p* < 0.01), skeletal muscle index (*p* < 0.001) and skeletal muscle density (*p* < 0.001). There was no significant association between hypoalbuminemia and either the presence of complications or their severity. In the absence of a systemic inflammatory response (*n* = 589), hypoalbuminemia was directly associated with MUST (*p* < 0.05) and inversely associated with BMI (*p* < 0.01), subcutaneous adiposity (*p* < 0.05), visceral adiposity (*p* < 0.05), skeletal muscle index (*p* < 0.01) and skeletal muscle density (*p* < 0.001). Hypoalbuminemia was, independently of inflammatory markers, associated with poorer cancer-specific and overall survival (both *p* < 0.001). The results suggest that hypoalbuminemia in patients with CRC reflects both increased nutritional risk and greater systemic inflammatory response and was independently associated with poorer survival in patients with CRC.

## 1. Introduction

Albumin is the most abundant plasma protein in humans, representing approximately 50% of the total protein content of plasma. Albumin has a myriad of important physiologic functions that are essential for normal health. In an adult, the normal range of serum albumin has been defined as 35–50 g/L and concentrations <35 g/L have been termed hypoalbuminemia [1]. The presence of hypoalbuminemia has long been associated with adverse clinical outcomes across a number of clinical specialties. For example, in a study of a large number of patients (*n* = 204,819) undergoing surgery, mainly for cancer (56%), Meyer et al. showed that hypoalbuminemia was associated with post-operative complications, increased length of hospital stay, and mortality. They recently concluded that there was a need to assess preoperative albumin levels and to manage nutritional status accordingly [2]. Some have concluded that this is antiquated thinking and ignores the fact that albumin is a negative acute phase reactant and that cancer can be considered as an inflammatory condition [3,4], and this may in part explain the prognostic value of albumin in patients with cancer. Also, serum albumin concentrations do not change in response to short-term changes in nutrient intake and in states of malnutrition [5,6,7].

In addition, it is clear that systemic inflammation reduces albumin concentrations independent of nutrient intake [5]. Indeed, the reduction in circulating albumin concentrations, as well as an increase in C-reactive protein, are linked with poor outcomes in patients with colorectal cancer independent of tumor stage and host factors [8,9,10]. Therefore, despite being addressed repeatedly in the literature, there is still a lack of quantitative data regarding the clinical significance of the factors that are associated with hypoalbuminemia (including nutritional risk, body composition and systemic inflammatory markers). Furthermore, little is known about whether hypoalbuminemia is associated with survival independent of these parameters. Through the use of Computerized tomography (CT scans), body composition can now be assessed as part of the routine clinical work up of patients with cancer, in particular in patients with operable disease [11]. The aim of the present study was to examine the relationship between hypoalbuminemia, nutrition-related risk, body composition, systemic inflammation and mortality in patients with operable colorectal cancer.

## 2. Patients and Methods

### 2.1. Patients

A retrospective cohort study from a prospective database of patients undergoing surgery for colorectal cancer (prospectively maintained as part of a surgical audit) was carried out. Consecutive patients who underwent potentially curative resection for colorectal cancer between March 2013 and June 2016 at Glasgow Royal Infirmary were studied. All patients with recorded routine laboratory measurements of white cells, neutrophil, lymphocyte, monocyte and platelet counts, albumin and C-reactive protein, and who had a preoperative CT scan, were included in the analysis. Patient comorbidity was classified using the American Society of Anesthesiologists (ASA) grading system [12]. Patients were classified as having anemia based on World Health Organization (WHO) guidelines for males (hemoglobin (Hb) < 130 g/L and females; Hb < 120 g/L) [13]. Patients were divided into normal and hypoalbuminemic groups according to the widely accepted definition of hypoalbuminemia (serum albumin > 35/< 35 g/L) [1]. A subgroup analysis for those patients without a systemic inflammatory response (CRP < 10 mg/L; *n* = 589) has been carried out to examine whether there is an independent effect of hypoalbuminemia on MUST and nutritional status based on BMI and body composition parameters. Any uncertainties of clinical outcome were addressed by a review of electronic and/or the physical case report.

The study had ethical approval in the UK (West of Scotland Ethics Committee: GN170N474) and was conducted in accordance with the Declaration of Helsinki. Furthermore, the study conformed to the Strengthening The Reporting of OBservational Studies in Epidemiolog (STROBE) guidelines for cohort studies.

### 2.2. Nutritional Risk

As part of routine clinical practice in the UK, MUST scores were recorded in patient notes within 24 h of admission. Clinical staff carried out the assessment (usually nursing staff).

MUST incorporates three components to determine the overall risk for malnutrition: current weight status using BMI, unintentional weight loss, and acute disease effect that has induced a phase of nil by mouth for > 5 days. Each parameter can be rated as 0, 1, or 2. Overall risk for malnutrition is established as low (score = 0), medium (score = 1), or high (score ≥ 2) [14].

### 2.3. Body Composition

CT images were obtained at the level of the third lumbar vertebra. Patients whose scans were taken 3 months or more prior to their surgery were excluded from the study. The median and range for the interval between CT scanning and operation was 0.91 months (0.03–2.83) prior to surgery. Scans with significant movement artefact or missing regions of interest were not considered for inclusion. Each image was analyzed using ImageJ software [15].

Region of interest (ROI) measurements were made of visceral fat (VFA), subcutaneous fat (SFA) and skeletal muscle areas (SMA) (cm^2^) using standard Hounsfield Unit (HU) ranges (adipose tissue −190 to −30, and skeletal muscle −29 to +150). These were then normalized for height^2^ to create indices; total fat index (TFI, cm^2^/m^2^), subcutaneous fat index (SFI, cm^2^/m^2^), visceral fat index (VFI, cm^2^/m^2^), and skeletal muscle index (SMI, cm^2^/m^2^). Skeletal muscle radiodensity (SMD, HU) was measured from the same ROI used to calculate SMI, as its mean HU.

As part of an ongoing research program in the Academic Unit of Surgery, training for body composition measurements and analysis was provided. The CT scan analysis was carried out by clinical research fellows (AA and RD). The inter-rater reliability was assessed in a sample of 40 patient images using interclass correlation coefficients (ICCC) (TFA ICCC = 1.000, SFA ICCC = 1.000, VFA ICCC = 1.000, SMA ICCC = 0.998, SMD ICCC = 0.972).

On the basis of CT-derived body composition, there have been a number of threshold values associated with survival in patients with cancer. The thresholds used for SMI and SMD were derived from large patient cohorts [16,17]. In particular, SMI is recognized to have prognostic value; the threshold used was derived from a large cohort of patients with operable colorectal cancer [16].

Visceral obesity was defined as VFA >160 cm^2^ for male patients and >80 cm^2^ for female patients [18]. High subcutaneous fat index (SFI) was defined as >50.0 cm^2^/m^2^ in males and >42.0 cm^2^/m^2^ in females [19]. Sarcopenia was described by Caan and colleagues as SMI < 52.3 cm^2^/m^2^ if BMI < 30 kg/m^2^ and SMI < 54.3 cm^2^/m^2^ if BMI > 30 kg/m^2^ in male patients and SMI < 38.6 cm^2^/m^2^ if BMI < 30 kg/m^2^ and an SMI < 46.6 cm^2^/m^2^ if BMI > 30 kg/m^2^ in female patients [16]. Myosteatosis was defined by SMD < 41HU in patients with BMI < 25 kg/m^2^ and <33HU in patients with BMI > 25 kg/m^2^ [17].

### 2.4. Inflammatory Markers

Prior to surgery, there was routine clinical assessment of the systemic inflammatory response using C-reactive protein, albumin and a differential white cell count. The clinical chemistry laboratory at Glasgow Royal Infirmary participated in external quality assurance/proficiency testing programs. Performance was acceptable throughout, which indicated that any changes in assay methodology did not result in any bias. An autoanalyzer was used to measure serum CRP (mg/L) and albumin (g/L) concentrations Grouping of the variables CRP, albumin, white cell, neutrophil, lymphocyte, monocyte and platelet counts was carried out using standard thresholds [20,21,22,23].

### 2.5. Statistical Analysis

Statistical analysis was performed using SPSS software, version 24.0 [24]. The data were manually extracted from the prospective dataset into SPSS. Categorical variables were analyzed using chi-square test χ2 test for linear-by-linear association, or χ2 test for 2-by-2 tables. Missing data were excluded from the analysis on a variable by variable basis.

Those with mortality within 30 days of the index procedure or during the index admission were excluded from subsequent survival analysis. The time between the date of surgery and the date of death of any cause was used to define overall survival (OS). The time between the date of surgery and the date of death of cancer was used to define cancer specific survival (CS). The rationale for including both survival end points was that nutritional biomarkers may impact on both end points. Survival data were analyzed using univariate and multivariate Cox regression. Those variables associated to a degree of *p* < 0.1 were entered into a backward conditional multivariate model. Two tailed *p* values < 0.001 were considered statistically significant.

### 2.6. Results

A total of 795 patients were included having undergone potentially curative surgery for colorectal cancer. The majority were male (55%) and less than 75 years old (74%), had low comorbidity (68%), were not anemic (60%), had a MUST score of 0 (79%) and were overweight or obese (66%). The majority of patients also had subcutaneous obesity (82%), visceral obesity (73%), low SMI (53%) and low SMD (63%) and did not have an elevated systemic inflammatory response as evidenced by white cell count (70%), neutrophil count (89%), monocyte count (88%), platelet count (88%) and C-reactive protein (75%).

The relationships between hypoalbuminemia and clinical and pathological characteristics such as body composition and host systemic inflammatory response are shown in Table 1.

In the presence of a systemic inflammatory response (*n* = 795), hypoalbuminemia was directly associated with greater age (*p* < 0.001), anemia (*p* < 0.001), comorbidity (*p* < 0.01), Tumor–Node–Metastasis TNM (*p* < 0.001) and MUST (*p* < 0.001) and inversely associated with BMI (*p* < 0.001), subcutaneous adiposity (*p* < 0.01), visceral obesity (*p* < 0.01), skeletal muscle index (*p* < 0.001) and skeletal muscle density (*p* < 0.001) and all components of a differential white cell count (all *p* < 0.001) except lymphocyte count (*p* = 0.074). There was no significant association between hypoalbuminemia and either the presence of complications or their severity. Hypoalbuminemia was directly associated with greater risk of cancer and non-cancer deaths (*p* < 0.001).

The relationships between hypoalbuminemia, clinical and pathological characteristics, and body composition are shown in Table 2. 

### 2.7. Survival

The relationship between hypoalbuminemia, nutrition status, body composition and cancer-specific survival in patients undergoing surgery for colorectal cancer is shown in Table 3.

On univariate analysis, age (*p* < 0.010), ASA (*p* < 0.05), TNM stage (*p* < 0.001), albumin (*p* < 0.001), anemia (*p* < 0.01), MUST score (*p* < 0.001) and BMI (*p* < 0.05) were significantly associated with cancer-specific survival. On multivariate analysis, TNM stage (HR = 3.42; 95% CI 2.16–5.40, *p* < 0.001) and MUST score (HR = 1.96; 95% CI 1.38–2.77, *p* = 0.007) were independently associated with cancer-specific survival. The relationships between hypoalbuminemia, systemic inflammation and cancer-specific survival in patients undergoing surgery for colorectal cancer is shown in Table 4.

On univariate analysis, albumin (*p* < 0.001) and all inflammatory markers (*p* < 0.05) were significantly associated with overall survival. On multivariate analysis, TNM stage (HR = 2.37; 95% CI 1.86–3.01, *p* = <0.001), albumin (HR = 2.58; 95% CI 1.78–3.73, *p* ≤ 0.001) and neutrophil count (HR = 1.79; 95% CI 1.11–2.87, *p* = 0.016) were independently associated with cancer-specific survival.

The relationship between hypoalbuminemia, nutrition status, body composition and overall survival in patients undergoing surgery for colorectal cancer is shown in Table 5.

On univariate analysis, age (*p* < 0.00), ASA (*p* < 0.001), TNM (*p* < 0.001), albumin (*p* < 0.001), anemia (*p* < 0.001), BMI (*p* < 0.001), MUST score (*p* < 0.001), subcutaneous adiposity (*p* < 0.050.021), SMI (*p* = 0.002) and SMD (*p* < 0.001) were significantly associated with overall survival. On multivariate analysis, age (HR = 1.65; 95% CI 1.20–2.26, *p* = 0.002), TNM (HR = 2.16; 95% CI 1.56–3.01, *p* < 0.001) and MUST score (HR = 1.53; 95% CI 1.14–2.05, *p* = 0.004), were independently associated with overall survival.

The relationship between hypoalbuminemia, systemic inflammation and overall survival in patients undergoing surgery for colorectal cancer is shown in Table 6.

On univariate analysis, albumin (*p* < 0.001) and all inflammatory markers (*p* < 0.01) were significantly associated with overall survival. On multivariate analysis, age (HR = 1.33; 95% CI 1.09–1.61, *p* < 0.01), ASA (HR = 1.43; 95% CI 1.18–1.72, *p* < 0.001), TNM stage (HR = 1.72; 95% CI 1.43–2.07, *p* < 0.001), albumin (HR = 1.95; 95% CI 1.45–2.62, *p* < 0.001), neutrophil count (HR = 1.59; 95% CI 1.08-2.33, *p* < 0.05) and monocyte count (HR = 1.57, 95% CI 1.07–2.29, *p* < 0.05) were independently associated with overall survival.

The Kaplan–Meier curves in Figure 1 show the relationship between albumin (normal vs. low) and cancer-specific survival (log rank *p* = 0.001), and Figure 2 shows the relationship between albumin (normal vs. low) and overall survival (log rank *p* = 0.001).

Figure 3 shows the relationship between albumin (normal: >35 g/L vs. low: <35 g/L) and cancer-specific survival (in the absence of a systemic inflammatory response, CRP < 10 mg/L) (log rank *p* = 0.001), and Figure 4 shows the relationship between albumin (normal: >35 g/L vs. low: <35 g/L) and overall survival (in the absence of a systemic inflammatory response, CRP < 10 mg/L) (log rank *p* = 0.001).

## 3. Discussion

The results of the present study show that hypoalbuminemia (<35 g/L) was associated with a number of factors including older age, anemia, greater comorbidity and more advanced tumor stage. Also, hypoalbuminemia was associated with nutritional risk as evidenced by MUST and body composition analysis. Furthermore, hypoalbuminemia was consistently associated with measurements of the systemic inflammatory response. Finally, hypoalbuminemia was associated with overall survival and cancer-specific survival independent of measurements of the systemic inflammatory response. Therefore, hypoalbuminemia objectively reflects both nutritional and inflammatory status and contributes prognostic value to markers of the systemic inflammatory response.

In the present study of patients with operable colorectal cancer, approximately 30% had hypoalbuminemia, 25% had evidence of a systemic inflammatory response and 50% had a low SMI prior to surgery. Therefore, even in early stage cancer, there is evidence of hypoalbuminemia, systemic inflammation and sarcopenia. Whilst the association between albumin and the systemic inflammatory response has been shown repeatedly in both acute and chronic inflammation [25,26], there is limited data on the association between albumin and body composition. With reference to hypoalbuminemia and low SMI, the present results are consistent with two small studies. Low muscle mass was associated with lower serum albumin concentration when body composition was measured using dual-energy X-ray absorptiometry in the elderly [27]. Similarly, there was an association between albumin concentrations and lower lean tissue as measured using a total body potassium scanner [28].

The basis of the association of hypoalbuminemia and low SMI is not clear. However, it might reflect the widespread association between the chronic activation of the systemic inflammatory response and catabolic loss of muscle protein in patients with cancer. Indeed, it has been recognized that proinflammatory cytokines such as interleukin-6 (IL-6), interleukin-1 (IL-1), tumor necrosis factor (TNF) and growth factor (GF), released as part of the systemic inflammatory response, have a profound catabolic effect on host tissue metabolism. For example, IL-6 stimulates liver production of CRP and other acute phase proteins, thus increasing the demand for amino acids against a background of limited intake. In particular, low circulating glutamine concentrations (the most abundant amino acid in plasma) have been reported to be associated with hypoalbuminemia, a low SMI (major source of glutamine) and poorer survival in patients with colorectal cancer [29,30,31,32]. Therefore, it would appear the most abundant circulating amino acid (glutamine), circulating protein (albumin) and body protein are linked in a chronic systemic inflammatory response and that albumin reflects the inter-organ amino acid flux during the chronic systemic inflammatory response.

The combination of C-reactive protein and albumin, termed the Glasgow Prognostic Score (GPS), has been reported to be informative of the nutritional risk of patients with cancer [33] and has been used successfully to predict survival in patients with a variety of common solid tumors, both in the operable [10] and advanced inoperable setting [9]. Therefore, taken together, the literature points to close linkage of nutritional risk and the systemic inflammatory response in patients with cancer and indicates that if one is measured, then the other should be measured also [7].

There are striking parallels with the work carried out examining micronutrient status in health and disease [34,35]. For example, there is now recognition that iron status should be interpreted in the context of inflammatory status [36]. Indeed, the combination of C-reactive protein and albumin has been used for this purpose. This perhaps points to the fundamental linkage of metabolism and inflammation in patients with cancer as there would appear to be an intimate evolutionary link between immune and metabolic responses in all mammalian cells, so called immunometabolism [37].

The implications of the present observations are important. It has become clear that the systemic inflammatory response is an important etiologic factor in the development of cancer cachexia [38,39]. However, the presence of albumin is not included in the criteria. Given the present and recent results from Silva et al. [33] it is clear that hypoalbuminemia in the presence of a systemic inflammatory response offers readily available additional insight into body composition and the likely outcome of the patient with operable colorectal cancer.

The main limitation of the present study was that it was a retrospective study of patients in a single institution. In particular, only patients with available CT scans were included in the analysis. However, given that a CT scan was part of the routine workup for operable colorectal cancer, less than 10% of patients had missing CT scans. In addition, the measurements were taken at one set point in time (for colorectal cancer staging) and therefore is a “snapshot” of the patient’s pathway and neither the point of onset of nutritional decline or changes in body composition (in particular skeletal muscle) are known. Therefore, there may be some selection bias in the present study. However, this study is, to our knowledge, the first to examine the association between a preoperative albumin, a nutritional risk tool (MUST), body composition, and systemic inflammation in a large number of patients undergoing surgery for primary operable cancer.

## 4. Conclusions

In summary, hypoalbuminemia was associated with greater nutritional risk, anemia, low skeletal muscle mass, low skeletal muscle density and the activation of the systemic inflammatory response. Moreover, it can provide useful independent prognostic value in patients with colorectal cancer.

## Figures and Tables

**Figure 1 cancers-12-01986-f001:**
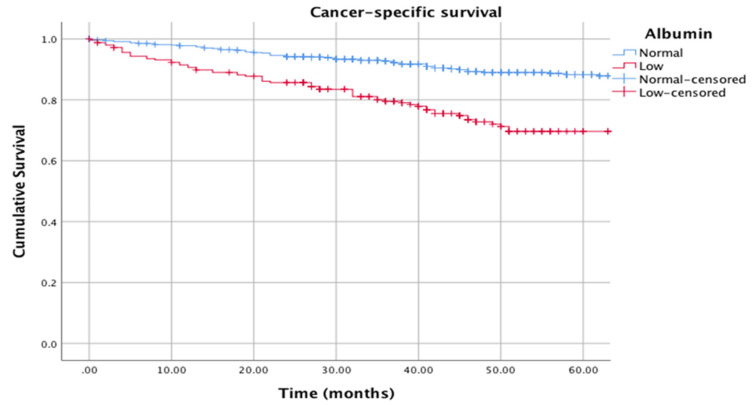
Kaplan–Meier curves showing the relationship between albumin (normal: >35 g/L vs. low: <35 g/L) and cancer-specific survival. *n* = 795. Median follow-up: 38 months. Log rank *p* = 0.001.

**Figure 2 cancers-12-01986-f002:**
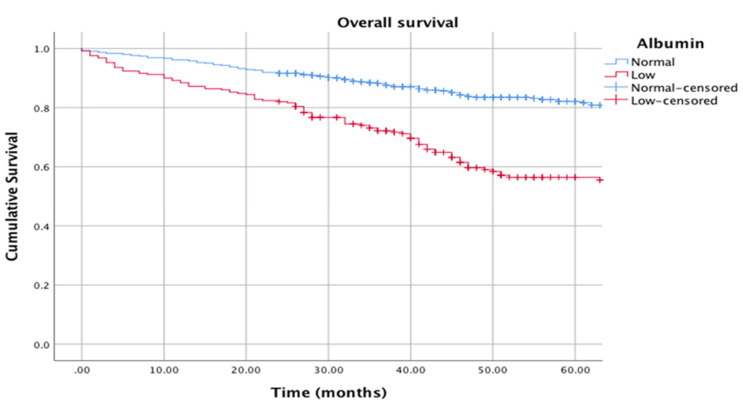
Kaplan–Meier curves showing the relationship between albumin (normal: >35 g/L vs. low: <35 g/L) and overall survival. *n* = 795. Median follow-up: 38 months. Log rank *p* = 0.001.

**Figure 3 cancers-12-01986-f003:**
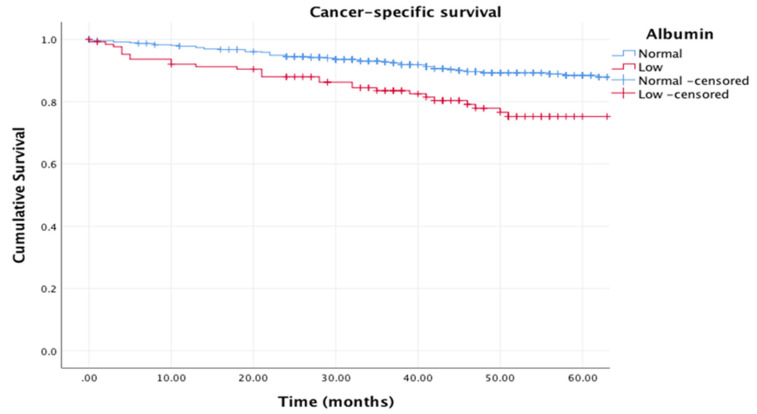
Kaplan–Meier curves showing the relationship between albumin (normal: >35 g/L vs. low: <35 g/L) and cancer-specific survival (in the absence of a systemic inflammatory response, CRP < 10 mg/L). *n* = 589. Median follow-up: 38 months. Log rank *p* = 0.001.

**Figure 4 cancers-12-01986-f004:**
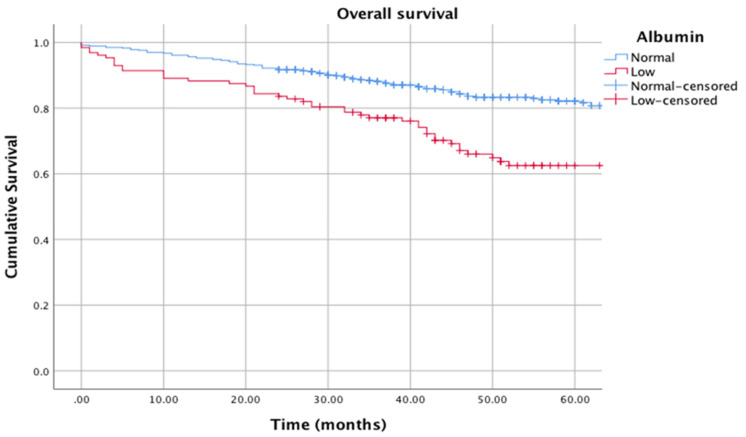
Kaplan–Meier curves showing the relationship between albumin (normal: >35 g/L vs. low: <35 g/L) and overall survival (in the absence of a systemic inflammatory response, CRP < 10 mg/L). *n* = 589. Median follow-up: 38 months. Log rank *p* = 0.001.

**Table 1 cancers-12-01986-t001:** The relationship between hypoalbuminemia, clinicopathological, CT-derived body composition and systemic inflammatory response in 795 patients undergoing surgery for colorectal cancer.

Characteristic		Total *n* = 795		
		Albumin > 35 g/L *n* = 545 (%)	Albumin < 35 g/L *n* = 250 (%)	*p*-Value ^a,b^
Clinicopathological				
Age	<65 y	221 (41)	64 (25)	**<0.001**
	65–74 y	225 (41)	79 (32)	
	>74 y	99 (18)	107 (43)	
Sex	Male	302 (55)	138 (55)	0.955
	Female	243 (45)	112 (45)	
ASA^c^	1	113 (25)	46 (18)	**0.004**
	2	241 (45)	108 (44)	
	3	147 (28)	78 (32)	
	4	11 (2)	14 (6)	
TNM^d^	0	15 (3)	3 (1)	**0.001**
	1	147 (28)	29 (12)	
	2	176 (33)	125 (50)	
	3	184 (34)	75 (30)	
	4	12 (2)	16 (7)	
**Nutrition Status and Body Composition**				
Hemoglobin	Normal	389 (72)	82 (33)	**<0.001**
	Anemia	155 (28)	165 (67)	
BMI (kg/m^2^)	Underweight	22 (4)	16 (7)	**<0.001**
	Normal weight	142 (27)	99 (40)	
	Overweight	177 (33)	76 (31)	
	Obese	191 (36)	54 (22)	
MUST score ^e^	0	217 (86)	59 (61)	**<0.001**
	1	13 (5)	18 (19)	
	2	23 (9)	20 (21)	
Subcutaneous adiposity	No	79 (16)	53 (24)	**0.010**
	Yes	421 (84)	170 (76)	
Visceral obesity	No	125 (24)	77 (33)	**0.013**
	Yes	397 (76)	159 (67)	
Low SMI ^f^	No	264 (53)	75 (34)	**<0.001**
	Yes	236 (47)	148 (66)	
Low SMD ^g^	No	223 (44)	49 (21)	**<0.001**
	Yes	288 (56)	183 (79)	
**Systemic Inflammatory Response**				
White cell count	<8.5 × 10^9-1^	406 (75)	153 (62)	**<0.001**
	8.5–11× 10^9-1^	106 (20)	60 (24)	
	>11 × 10^9-1^	31 (6)	34 (14)	
Lymphocyte count	>3.0 × 10^9-1^	29 (5)	7 (3)	0.074
	1.0–3.0 × 10^9-1^	450 (83)	205 (82)	
	<1.0 × 10^9-1^	66 (12)	38 (15)	
Neutrophil count	<7.5 × 10^9-1^	506 (93)	205 (82)	**<0.001**
	>7.5 × 10^9-1^	39 (7)	45 (18)	
Monocyte count	<0.9 × 10^9-1^	495 (92)	205 (84)	**0.002**
	>0.9 × 10^9-1^	46 (8)	39 (16)	
Platelet count	<400 × 10^9-1^	506 (93)	193 (78)	**<0.001**
	>400 × 10^9-1^	36 (7)	54 (22)	
CRP ^h^	<10 mg/L	461 (85)	128 (51)	**<0.001**
	>10 mg/L	79 (15)	121 (49)	
**Clinical Outcome**				
Non-infective complication	No	447 (83)	191 (78)	0.068
	Yes	95 (18)	55 (22)	
Infective complication	No	394 (73)	168 (68)	0.206
	Yes	148 (27)	78 (32)	
Clavien Dindo grade	0	337 (62)	142 (58)	0.309
	1–2	154 (28)	80 (32)	
	3–5	50 (9)	24 (10)	
5-year survival	Alive	432 (81)	140 (57)	**<0.001**
	Cancer-specific death	57 (11)	66 (27)	
	Non-cancer death	45 (8)	38 (16)	

^a^ Chi test for linear by linear association; ^b^ Significant values in bold *p* < 0.05; ^c^ ASA: American Society of Anesthesiologists; ^d^ TNM: Tumor–Node–Metastasis; ^e^ MUST: Malnutrition Universal Screening Tool. Overall risk for malnutrition is established as low (score = 0), medium (score = 1), or high (score ≥2); ^f^ SMI: Skeletal Muscle Index; ^g^ SMD: Skeletal Muscle Density; ^h^ CRP: C-Reactive Protein.

**Table 2 cancers-12-01986-t002:** The relationship between hypoalbuminemia, clinicopathological, CT-derived body composition and systemic inflammatory response in 589 patients with CRP <10 mg/L undergoing surgery for colorectal cancer.

Characteristic		Total *n* = 589		
		Albumin > 35 g/L *n*= 461(%)	Albumin < 35 g/L *n* = 128 (%)	*p*-Value ^a,b^
Clinicopathological				
Age	<65 y	185 (40)	31 (24)	**<0.001**
	65–74 y	187 ((41)	42 (33)	
	>74 y	89 (19)	55 (43)	
Sex	Male	256 (55)	70 (55)	0.865
	Female	205 (45)	58 (45)	
ASA^c^	1	118 (26)	30 (24)	0.094
	2	201 (45)	49 (40)	
	3	124 (27)	39 (31)	
	4	8 (2)	7 (6)	
TNM ^d^	0	14 (3)	2 (2)	**0.024**
	1	132 (29)	22 (17)	
	2	142 (32)	54 (43)	
	3	151 (34)	42 (33)	
	4	11 (2)	7 (5)	
**Nutrition Status and Body Composition**				
Hemoglobin	Normal	332 (72)	48 (38)	**<0.001**
	Anemia	128 (28)	77 (62)	
BMI (kg/m^2^)	Underweight	18 (4)	5 (4)	**0.009**
	Normal weight	121 (27)	52 (41)	
	Over weight	157 (35)	37 (29)	
	Obese	154 (34)	32 (25)	
MUST score ^e^	0: No risk	186 (84)	36 (69)	**0.016**
	1 and 2: medium and high risk	36 (16)	16 (31)	
Subcutaneous adiposity	No	64 (15)	28 (25)	0.016
	Yes	362 (85)	86 (75)	
Visceral obesity	No	104 (23)	39 (33)	0.036
	Yes	341 (77)	80 (67)	
Low SMI ^f^	No	227 (53)	44 (37)	**0.005**
	Yes	199 (47)	70 (61)	
Low SMD ^g^	No	192 (44)	32 (27)	**0.001**
	Yes	243 (56)	85 (73)	
**Systemic Inflammatory Response**				
White cell count	<8.5 × 10^9-1^	354 (77)	94 (75)	0.853
	8.5–11× 10^9-1^	86 (19)	27 (22)	
	>11 × 10^9-1^	19 (4)	4 (3)	
Lymphocyte count	>3.0 × 10^9-1^	26 (6)	3 (2)	0.079
	1.0–3.0 × 10^9-1^	378 (82)	104 (81)	
	<1.0 × 10^9-1^	57 (12)	21 (16)	
Neutrophil count	<7.5 × 10^9-1^	434 (94)	116 (91)	0.157
	>7.5 × 10^9-1^	27 (6)	12 (9)	
Monocyte count	<0.9 × 10^9-1^	424 (93)	112 (90)	0.243
	>0.9 × 10^9-1^	33 (7)	13 (10)	
Platelet count	<400 × 10^9-1^	434 (95)	116 (92)	0.253
	>400 × 10^9-1^	24 (5)	10 (8)	
**Clinical Outcome**				
Non-infective complication	No	381 (83)	98 (78)	0.162
	Yes	77 (17)	28 (22)	
Infective complication	No	337 (74)	90 (71)	0.630
	Yes	121 (26)	36 (29)	
Clavien Dindo grade	0	287 (63)	76 (60)	0.707
	1–2	122 (27)	37 (29)	
	3–5	46 (10)	13 (11)	
5-year survival	Alive	369 (80)	79 (62)	**<0.001**
	Cancer-specific death	52 (11)	31 (24)	
	Non-cancer death	40 (9)	18 (14)	

^a^ Chi test for linear by linear association; ^b^ Significant values in bold *p* < 0.05; ^c^ ASA: American Society of Anesthesiologists; ^d^ TNM: Tumor**–**Node**–**Metastasis; ^e^ MUST: Malnutrition Universal Screening Tool. Overall risk for malnutrition is established as low (score = 0), medium (score = 1), or high (score ≥2); ^f^ SMI: Skeletal Muscle Index; ^g^ SMD: Skeletal Muscle Density.In the absence of a systemic inflammatory response (C-reactive protein < 10 mg/L, *n* = 589), hypoalbuminemia was directly associated with greater age (*p* < 0.001), anemia (*p* < 0.001), TNM (*p* < 0.05) and MUST (*p* < 0.05) and inversely associated with BMI (*p* < 0.01), subcutaneous adiposity (*p* < 0.05), visceral adiposity (*p* < 0.05), skeletal muscle index (*p* < 0.01) and skeletal muscle density (*p* < 0.001), but not components of a differential white cell count. There was also no significant association between hypoalbuminemia and either the presence of complications or their severity. Hypoalbuminemia was directly associated with greater risk of cancer and non-cancer deaths (*p* < 0.001).

**Table 3 cancers-12-01986-t003:** The relationship between hypoalbuminemia, nutritional risk, CT-derived body composition and cancer-specific survival in 795 patients undergoing surgery for colorectal cancer.

Variable ^a^	Cancer-Specific Survival			
	Univariate HR (95% CI)*	*p*-Value	Multivariate HR (95% CI)	*p*-Value
Age (<65/ 65–74/ >74) y	1.35 (1.07–1.70)	**0.010**	1.05 (0.69–1.62)	0.152
ASA grade ^b^ (1/2/3/4)	1.29 (1.03–1.63)	**0.025**	1.31 (0.86–1.98)	0.145
TNM stage (0/1/2/3/4)	2.52 (1.98–3.20)	**<0.001**	3.42 (2.16–5.40)	**<0.001**
Hypoalbuminemia (no/yes)	2.93 (2.06–4.18)	**<0.001**	1.68 (0.84–3.35)	0.059
Anemia (no/yes)	1.64 (1.15–2.34)	**0.006**	1.24 (0.65–2.35)	0.192
BMI (kg/m^2^) (underweight/normal/overweight/obese)	0.81 (0.66–0.99)	**0.042**	0.88 (0.61–1.26)	0.588
MUST score ^c^ (0/1/2)	1.79 (1.27–2.51)	**0.001**	1.96 (1.38–2.77)	**<0.001**
Subcutaneous adiposity (no/yes)	0.79 (0.49–1.26)	0.328	-	-
Visceral obesity (no/yes)	0.94 (0.62–1.43)	0.796	-	-
Low SMI ^d^ (no/yes)	1.20 (0.82–1.80)	0.326	-	-
Low SMD ^c^ (no/yes)	1.38 (0.92–2.07)	0.118	-	-

* HR: Hazard ratio, CI: Confidence Interval. ^a^ Variables associated to a degree of *p* < 0.1 were entered into a backward conditional multivariate model. Two-tailed *p* values <0.001 were considered statistically significant; ^b^ ASA: American Society of Anesthesiologists; ^c^ MUST: Malnutrition Universal Screening Tool. Overall risk for malnutrition is established as low (score = 0), medium (score = 1), or high (score ≥2); ^d^ SMI: Skeletal Muscle Index; ^e^ SMD: Skeletal Muscle Density.

**Table 4 cancers-12-01986-t004:** The relationship between hypoalbuminemia, systemic inflammatory response and cancer-specific survival in 795 patients undergoing surgery for colorectal cancer.

Variable ^a^	Cancer-Specific Survival			
	Univariate HR (95% CI)*	*p*-Value	Multivariate HR (95% CI)	*p*-Value
Age (<65/ 65–74/ >74) y	1.35 (1.07–1.70)	**0.010**	1.14 (0.89–1.46)	0.164
ASA grade (1/2/3/4)	1.29 (1.03–1.63)	**0.025**	1.21 (0.95–1.55)	0.072
TNM stage (0/1/2/3/4)	2.52 (1.98–3.20)	**<0.001**	2.37 (1.86–3.01)	**<0.001**
Hypoalbuminemia (no/yes)	2.93 (2.06–4.18)	**<0.001**	2.58 (1.78–3.73)	**<0.001**
CRP (normal/abnormal)	1.94 (1.34–2.80)	**<0.001**	1.21 (0.79–1.84)	0.442
White cell count (normal/abnormal)	1.55 (1.21–1.99)	**<0.001**	1.03 (0.71–1.50)	0.719
Neutrophil count (normal/abnormal)	2.62 (1.69–4.08)	**<0.001**	1.79 (1.11–2.87)	**0.016**
Monocyte count (normal/abnormal)	2.36 (1.50–3.73)	**<0.001**	1.67 (1.04–2.70)	0.167
Platelet count (normal/abnormal)	1.69 (1.05–2.74)	**0.030**	0.85 (0.48–1.49)	0.944

* HR: Hazard ratio, CI: Confidence Interval ^a^ Variables associated to a degree of *p* < 0.1 were entered into a backward conditional multivariate model. Two-tailed *p* values <0.001 were considered statistically significant.

**Table 5 cancers-12-01986-t005:** The relationship between hypoalbuminemia, nutritional risk, CT-derived body composition and overall survival in 795 patients undergoing surgery for colorectal cancer.

Variable ^a^	Overall Survival			
	Univariate HR (95% CI)*	*p*-Value	Multivariate HR (95% CI)	*p*-Value
Age (<65/ 65–74/ >74) y	1.63 (1.36–1.95)	**<0.001**	1.65 (1.20–2.26)	**0.002**
ASA grade ^b^ (1/2/3/4)	1.61 (1.35–1.93)	**<0.001**	1.39 (0.99–1.93)	0.093
TNM stage (0/1/2/3/4)	1.77 (1.48–2.11)	**<0.001**	2.16 (1.56–3.01)	**<0.001**
Hypoalbuminemia (no/yes)	2.62 (1.99–3.45)	**<0.001**	1.52 (0.87–2.66)	0.112
Anemia (no/yes)	1.74 (1.32–2.30)	**<0.001**	1.16 (0.69–1.94)	0.290
BMI (kg/m^2^) (underweight/normal/overweight/obese)	0.77 (0.66–0.90)	**0.001**	0.87 (0.65–1.16)	0.627
MUST score ^c^ (0/1/2)	1.63 (1.24–2.14)	**<0.001**	1.53 (1.14–2.05)	**0.004**
Subcutaneous adiposity (no/yes)	0.66 (0.47–0.94)	**0.021**	-	-
Visceral obesity (no/yes)	0.87 (0.63–1.18)	0.377	-	-
Low SMI ^d^ (no/yes)	1.6 (1.19–2.23)	**0.002**	-	-
Low SMD ^c^ (no/yes)	1.79 (1.30–2.48)	**<0.001**	1.14 (0.60–2.15)	0.629

* HR: Hazard ratio, CI: Confidence Interval ^a^ Variables associated to a degree of *p* < 0.1 were entered into a backward conditional multivariate model. Two-tailed *p* values <0.001 were considered statistically significant; ^b^ ASA: American Society of Anesthesiologists; ^c^ MUST: Malnutrition Universal Screening Tool. Overall risk for malnutrition is established as low (score = 0), medium (score = 1), or high (score ≥2).; ^d^ SMI: Skeletal Muscle Index; ^e^ SMD: Skeletal Muscle Density.

**Table 6 cancers-12-01986-t006:** The relationship between hypoalbuminemia, systemic inflammation, and overall survival in in 795 patients undergoing surgery for colorectal cancer.

Variable ^a^	Overall Survival			
	Univariate HR (95% CI)*	*p*-Value	Multivariate HR (95% CI)	*p*-Value
Age (<65/ 65–74/ >74) y	1.63 (1.36–1.95)	**<0.001**	1.33 (1.09–1.61)	**0.004**
ASA grade (1/2/3/4)	1.61 (1.35–1.93)	**<0.001**	1.43 (1.18–1.72)	**<0.001**
TNM stage (0/1/2/3/4)	1.77 (1.48–2.11)	**<0.001**	1.72 (1.43–2.07)	**<0.001**
Hypoalbuminemia (no/yes)	2.62 (1.99–3.45)	**<0.001**	1.95 (1.45–2.62)	**<0.001**
CRP (normal/abnormal)	1.85 (1.39–2.46)	**<0.001**	1.167 (0.83–1.62)	0.318
White cell count (normal/abnormal)	1.53 (1.26–1.87)	**<0.001**	0.98 (0.73–1.32)	0.897
Neutrophil count (normal/abnormal)	2.29 (1.60–3.28)	**<0.001**	1.59 (1.08–2.33)	**0.017**
Monocyte count (normal/abnormal)	2.19 (1.52–3.14)	**<0.001**	1.57 (1.07–2.29)	**0.019**
Platelet count (normal/abnormal)	1.73 (1.19–2.52)	**0.004**	1.05 (0.69–1.61)	0.628

* HR: Hazard ratio, CI: Confidence Interval ^a^ Variables associated to a degree of *p* < 0.1 were entered into a backward conditional multivariate model. Two-tailed *p* values <0.001 were considered statistically significant.

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
