# Peer review of "Hypoalbuminemia Reflects Nutritional Risk, Body Composition and Systemic Inflammation and Is Independently Associated with Survival in Patients with Colorectal Cancer"

_cancers, 2020, doi:10.3390/cancers12071986_

Round 1

Reviewer 1 Report

This paper examines the association between hypoalbuminemia, nutritional risk, body composition and systemic inflammation in undergoing surgery for primary operable cancer. The most striking finding is the description of the relationship between SMI and hypoalbuminemia which has not been widely reported previously and value of using albumin measurements as a prognostic marker.

Abstract

Line 15 GLIM criteria: please explain

Lind 26: include type of statistical analysis in methods section of abstract

Word count of abstract: please check and edit to 250-300 words. Suggest highlight key results especially associated with SMI

Methods

Line 90: was consent obtained from patients at the time of data collection?

Line 135: include version of SPSS

Line 142: some analysis has been carried out on a subset of the population with CRP<10. Please include rationale for this in methods.

Line 142: Please describe the outcome variables used in the multivariate regression; give definition of cancer survival and overall survival and rationale for using both types of models

Results

Table 1: please write out acronyms for TNM; Table 2 add CRP and TNM to acronymns list and write out

Discussion

Line 32: include which survival referring to: both cancer and overall?

Line 77 study limited as includes patients who operation with available CT scans; thus, there may be a selection bias regarding patients excluded without certain data collected;

Reviewer 2 Report

this manuscript was well written and aim to find a corrleation between nutirional index, body compostion and systmeic inflammatory reaction  and survival in colorectal cancer.

I would like to address a couple of issues.

  1. Title of this manuscript seesm to be not adequate and too broad to difficult to understand, Therefore the title of this paper be changed toward to the more sceintific tredn,
  2. As author mentioned, although nitritional index, systemic inflammaroty index were well knowm prognositc factors in CRC patients, but this study was doen only one time check before treatment, So this is the the most limitation of this retrospective study,
  3. Although the nutironal index and MUST are the significant prognostic factors in CRC, as you know TNM stage is the most strong prognostic factor, therefore, can you analyzed the nutritioanl and systemic inlfammatory factor accroding each stage of CRC.
  4. Tabel and Figuer shoud be be more clear,
  5. The result section should be more summarized step by step and more clearly explained.

Reviewer 3 Report

The document has a good level for the journal. The article analyzes relevant objectives with an adequate statistical and methodological design. The slight improvements that I could recommend refer to personal writing styles that are not related to the quality of the document. Therefore, I do not consider it necessary to refer them. Congratulations on the publication. It can be published in its current format.

Author Response

The authors acknowledge the supportive comments of the reviewer.

Round 2

Reviewer 2 Report

I appreciate author response according to reviewer comments sincerely,